



# Divergent historical GPP trends among state-of-the-art multi-model simulations and satellite-based products

Ruqi Yang[1,2], Jun Wang[3,4*], Ning Zeng[1,5], Stephen Sitch[6], Wenhan Tang[1,2], Matthew Joseph McGrath[7], Qixiang Cai[1], Di Liu[1], Danica Lombardozzi[8], Hanqin Tian[9], Atul K Jain[10], Pengfei Han[1,11]

[1]State Key Laboratory of Numerical Modeling for Atmospheric Sciences and Geophysical Fluid Dynamics, Institute of Atmospheric Physics, Chinese Academy of Sciences, Beijing, China
[2]College of Earth and Planetary Sciences, University of Chinese Academy of Sciences, Beijing 100049, China
[3]International Institute for Earth System Science, Nanjing University, Nanjing, Jiangsu 210023, China
[4]Jiangsu Provincial Key Laboratory of Geographic Information Science and Technology, Key Laboratory for Land Satellite
Remote Sensing Applications of Ministry of Natural Resources, School of Geography and Ocean Science, Nanjing University, Nanjing, Jiangsu 210023, China
[5]Department of Atmospheric and Oceanic Science, and Earth System Science Interdisciplinary Center, University of Maryland, College Park, Maryland, USA
[6]College of Life and Environmental Sciences, University of Exeter, UK.
[7]Laboratoire des Sciences du Climat et de l'Environnement/Institut Pierre Simon Laplace, Commissariat à l'Énergie, Atomique et aux Énergies Alternatives–CNRS–Université de Versailles Saint-Quentin, Université Paris-Saclay, F-91191 Gif-sur-Yvette, France
[8]National Center for Atmospheric Research, Climate and Global Dynamics Laboratory, Boulder, CO, USA
[9]International Center for Climate and Global Change Research, School of Forestry and Wildlife Sciences, Auburn
University, Auburn, AL 36849, USA.
[10]Department of Atmospheric Science, University of Illinois, Urbana, IL 61801, USA
[11]Carbon Neutrality Research Center, Institute of Atmospheric Physics, Chinese Academy of Sciences, Beijing, China

*Correspondence to*: Jun Wang (wangjun@nju.edu.cn)

**Abstract.** Understanding historical changes in gross primary productivity (GPP) is essential for better predicting the future
global carbon cycle. However, the historical trends of terrestrial GPP, owing to the $CO_2$ fertilization effect, climate, and land-use change, remain largely uncertain. Using long-term satellite-based near-infrared radiance of vegetation (NIRv), a proxy for GPP, and multiple GPP datasets derived from satellite-based products, Dynamic Global Vegetation Model (DGVM) simulations, and machine learning techniques, here we comprehensively investigated their trends and analyzed the causes for any discrepancies during 1982–2015. Although spatial patterns of climatological annual GPP from all products and NIRv are
highly correlated ($r > 0.84$), the spatial correlation coefficients of trends between DGVM GPP and NIRv significantly decreased (with the ensemble mean of $r = 0.49$) and even the spatial correlation coefficients of trends between other GPP products and NIRv became negative. By separating the global land into the tropics plus extra-tropical southern hemisphere (Trop+SH) and extra-tropical northern hemisphere (NH), we found that, during 1982–2015, simulated GPP from most of the models showed a stronger increasing trend over Trop+SH than NH. In contrast, the satellite-based GPP products indicated a
substantial increase over NH. Mechanistically, model sensitivity experiments indicated that the increase of annual GPP was dominated by the $CO_2$ fertilization effect (Global: 83.9%), albeit a large uncertainty in magnitude among individual simulations. However, the spatial distribution of inter-model spreads of GPP trends resulted mainly from climate and land-use change rather



than CO₂ fertilization effect. Trends after 2000 were different from the full time-series, showing that satellite-based GPP products suggested weakened rising trends over NH and even significantly decreasing trends over Trop+SH, while the trends

from DGVMs kept increasing. The inconsistencies are very likely caused by the contrasting performances between satellite-derived and DGVM simulated vegetation structure parameter (leaf area index, LAI). Therefore, the uncertainty in satellite-based GPP products induced by highly uncertain LAI data in the tropics undermines their roles in assessing the performance of DGVM simulations and understanding the changes of global carbon sinks.

## 1 Introduction

The gross primary productivity (GPP) and the ecosystem respiration (ER) dominate carbon fluxes from terrestrial ecosystems. Therefore, quantifying global terrestrial GPP is essential to understanding the global carbon cycle (Ryu et al., 2019). To date, there are multiple global GPP products, mainly including the up-scaled products from the eddy covariance flux data by machine learning techniques (Beer et al., 2010; Jung et al., 2020), satellite-based estimates by light-use efficiency (LUE) model (Running et al., 2004; Yuan et al., 2010; Joiner et al., 2018; Zheng et al., 2020), and simulations by the state-of-the-art Dynamic

Global Vegetation Models (DGVMs) (Huntzinger et al., 2013; Sitch et al., 2015).

The machine learning FLUXCOM GPP products based on the global FLUXNET network, remote sensing, and meteorological input (Jung et al., 2020; Pastorello et al., 2020) are widely used in terrestrial carbon cycle studies. Taking FLUXCOM GPP as a benchmark, research has explored the interannual variation, seasonal cycle, and climatology pattern of global and regional

GPP (Chen et al., 2017; Jia et al., 2020; Zhang and Ye, 2021). However, due to the lack of the CO₂ fertilization effect, the performance of this product on the long-term GPP trend is not realistic (Jung et al., 2020). Based on the LUE principle and derived from the Advanced Very High-Resolution Radiometer (AVHRR) and the Moderate Resolution Imaging Spectroradiometer (MODIS) datasets, the satellite-based GPP estimates include MOD17, GLASS, GIMMS, FluxSat, WECANN, and revised EC_LUE GPP product (Running et al., 2004; Yuan et al., 2007; Smith et al., 2016; Alemohammad et

al., 2017; Joiner et al., 2018; Zheng et al., 2020). These GPP products capture the seasonal variation, spatial distribution, and interannual variation to a large extent (Wang et al., 2014), but do not always account for the CO₂ fertilization effect (O'sullivan et al., 2020). For DGVM simulations, different forcing datasets, parameterizations, and processes considered can make the surprising differences in model representation of responses of photosynthesis to CO₂ concentration, soil moisture, temperature, and water vapor deficit (Rogers, 2014; Rogers et al., 2017). These differences caused large inter-model spreads in GPP

simulations (Ito et al., 2017). Hence, many efforts have been made to constrain the global GPP magnitude based on the satellite observations like solar-induced chlorophyll fluorescence (SIF) (Hashimoto et al., 2013; Macbean et al., 2018; Bacour et al., 2019; Norton et al., 2019; Wang et al., 2021a).



The application of satellite-derived GPP proxy datasets provides a breakthrough for estimating global GPP (Running et al.,
2004; Badgley et al., 2019; Piao et al., 2020). Many GPP proxy indices, such as normalized difference vegetation index (NDVI),
enhanced vegetation index (EVI), and SIF, have been widely used to estimate the global GPP (Frankenberg et al., 2011;
Guanter et al., 2014). However, each of them has its shortcomings. For example, NDVI can be saturated in tropical regions,
demonstrating its nonlinear relationship with GPP (Badgley et al., 2017; Badgley et al., 2019; Camps-Valls et al., 2021). The
EVI index improves the NDVI algorithm, but this index has not entirely solved the saturation problem (Huete et al., 2002).
Without dealing with the problem of distinguishing whether the signal comes from the plant or other interference factors,
satellite retrieval of SIF measures the light emitted by chlorophyll in leaves and can be used as a robust proxy of GPP
(Frankenberg et al., 2011; Mohammed et al., 2019). However, the time range of global SIF products is short, with direct
observations only available from 2007. Representing the proportion of reflected near-infrared radiation attributable to
vegetation, NIRv is a relatively recent GPP proxy (Badgley et al., 2017). Compared to NDVI and EVI, the saturation problem
of NIRv and GPP in the tropical region is weakened because of the mixed effects of background brightness, leaf area, and the
distribution of canopy photosynthetic capacity with depth were largely eliminated. Since NIRv can be directly obtained from
observational datasets of the AVHRR sensors, it can be derived from 1982 to the present. Moreover, previous studies have
shown that NIRv and SIF are closely related and indicated that NIRv could well represent changes in GPP (Badgley et al.,
2017; Badgley et al., 2019; Camps-Valls et al., 2021; Wang et al., 2021c).

Although there have been a lot of studies focusing on extreme anomalies, the seasonal cycle, interannual variation, and the
climatological pattern of global and regional GPP based on the multiple GPP products and proxy indices (Chen et al., 2017;
Madani et al., 2020; Wang et al., 2021b), few efforts have been devoted to evaluate the long-term GPP trends across different
GPP sources and to analyze the causes of uncertainties. This study comprehensively investigates historical GPP trends during
1982−2015, based on the satellite-derived GPP proxy (NIRv), TRENDYv6 multi-model simulations, machine-learning
products, satellite-based estimates, and site-level observations. Section 2 describes the datasets and statistical methods used.
The comparison of GPP trends among DGVM simulations and satellite-based GPP products is in section 3.1. The mechanisms
of the trend attributions are proposed and explained in section 3.2. The discussions about the uncertainties in GPP trends are
included in section 3.3. Finally, the main conclusions of the results are summarized in section 4.

## 2 Datasets and methods

### 2.1 TRENDYv6 multi-model simulated GPP

We used the model simulation results conducted under the auspices of the "Trends and drivers of the regional scale sources
and sinks of carbon dioxide" (TRENDY) Project (Sitch et al., 2015). We used 10 DGVMs in the TRENDYv6 project for the
period of 1982-2015, including CABLE (Haverd et al., 2018), CLASS-CTEM (Melton and Arora, 2016), CLM4.5 (Oleson et





al., 2010), DLEM (Tian et al., 2015), ISAM (Jain et al., 2013), OCN (Zaehle et al., 2010), ORCHIDEE-MICT (Guimberteau et al., 2018), ORCHIDEE (Krinner et al., 2005), VEGAS (Zeng et al., 2005), and VISIT (Kato et al., 2013). There is a suite of experimental protocols in the TRENDY project, and we here explored GPP trends and their mechanisms using the GPP outputs from three simulations. In detail, DGVMs were run under the varying $CO_2$ concentration, and constant climate conditions and

land-use change in S1; the varying $CO_2$ concentration and climate conditions, with constant land-use change in S2; the varying $CO_2$ concentration, climate conditions, and land-use change in S3. Hence, the S1 scenario represents the impact of the $CO_2$ fertilization effect. The contributions of climate change and land-use change (hereafter "LUC") are calculated through the differences between S2 and S1, S3 and S2, respectively. These modelling details are listed in Table 1.

In sections 3.2 and 3.3.1, we calculated the ensemble mean of the 7 model simulations, which included all scenarios as the DGVM ensemble GPP and calculated their standard deviation to represent inter-model spread across these models. In other sections which only need results from S3, we used the ensemble mean simulations from 10 models.

**Table 1. Information of TRENDYv6 models used in this study. S1 represents the impact of the $CO_2$ fertilization effect, S2 represents**
**the impact of the $CO_2$ fertilization effect and climate change, S3 represents the impact of the $CO_2$ fertilization effect, climate change, and LUC.**

| Models | Spatial resolution | S1[a] | S2 | S3 | References |
|---|---|---|---|---|---|
| **CABLE** | 0.5º × 0.5º | √ | | √ | Haverd et al., 2018 |
| **CLASS-CTEM** | T42 | √ | √ | √ | Melton and Arora 2016 |
| **CLM4.5** | 0.94º × 1.25º | √ | √ | √ | Oleson et al., 2010 |
| **DLEM** | 0.5º × 0.5º | √ | √ | √ | Tian et al. 2015 |
| **ISAM** | 0.5º × 0.5º | √ | √ | √ | Jain et al., 2013 |
| **OCN** | 0.5º × 0.5º | | | √ | Zaehle and Friend 2010 |
| **ORCHIDEE-MICT** | 1º × 1º | | | √ | Guimberteau et al., 2018 |
| **ORCHIDEE** | 0.5º × 0.5º | √ | √ | √ | Krinner et al., 2005 |
| **VEGAS** | 0.5º × 0.5º | √ | √ | √ | Zeng et al., 2005 |
| **VISIT** | 0.5º × 0.5º | √ | √ | √ | Ito and Inatomi., 2011 |

[a]Simulation datasets in the corresponding experiments (S1, S2, and S3) as available for models indicated with the notation of "√".

## 2. 2 FLUXCOM GPP

The FLUXCOM datasets comprised of 120 global carbon flux products generated by nine machine learning techniques, based on site-level observed GPP measured by eddy covariance and upscaled with remote sensing information and meteorology data (Jung et al., 2020). This research used the ensemble mean of GPP datasets forced by CRUJRA climate data and generated



from three machine learning techniques (random forest, artificial neural network, and multivariate adaptive regression splines) from 1982 to 2015. The original spatial resolution of this dataset is 0.5 º × 0.5 º.


## 2.3 Satellite-based GPP products

In this study, the GLASS GPP and revised EC-LUE GPP estimates were used as representatives of long-term satellite-based GPP products from 1982 to 2015.

GLASS GPP originated from the Eddy Covariance–Light Use Efficiency (EC-LUE) model (Yuan et al., 2007), which considered various impact factors (NDVI, photosynthetically activate radiation, temperature, $CO_2$ concentrations, the Bowen ratio of sensible to latent heat flux, water vapor pressure deficit, direct radiation fluxes, and diffuse radiation fluxes) and nine ecosystem types to accurately estimate the long-term change of GPP (Yuan et al. 2019).  The original spatial resolution of this dataset is 0.05 º × 0.05 º.


The revised EC-LUE GPP is a long-term GPP dataset based on the LUE equation. Zheng et al. (2020) generated the revised EC-LUE GPP using the following formula:

$$GPP = (\varepsilon_{msu} \times APARsu + \varepsilon_{msh} \times APAR_{sh}) \times Cs \times min(Ts, Ws) \tag{1}$$

where $\varepsilon_{msu}$ is the maximum LUE of sunlit leaves and $\varepsilon_{msh}$ is the maximum LUE of shaded leaves; $APARsu$ is the PAR
absorbed by sunlit leaves and $APAR_{sh}$ is the PAR absorbed by shaded leaves. $Cs, Ts$ , and $Ws$ represent the downward regulation scalars of atmospheric $CO_2$ concentration, temperature, and VPD on LUE with the range from 0 to 1. Specifically, the direct effect of $CO_2$ fertilization on GPP is determined by the following equations:

$$C_s = \frac{C_i - \varphi}{C_i + 2\varphi} \tag{2}$$

$$C_i = C_a \times \chi \tag{3}$$

where $C_i$ represents the $CO_2$ concentration inside the leaf, $C_a$ is the atmospheric $CO_2$ concentration, $\varphi$ means the $CO_2$ compensation point in the absence of dark respiration (ppm), and $\chi$ means the ratio of $CO_2$ concentration inside the leaf to that in the atmosphere (Farquhar et al., 1980).

After adding the effect of $CO_2$ fertilization, the GPP generated from this model is closer to the site observation data of more
than five years ($R^2$ = 0.44) than that of other LUE models ($R^2$ ranged from 0.06 to 0.30) (Zheng et al., 2020). The original spatial resolution of this dataset is 0.05 º × 0.05 º.

The spatial pattern and temporal changes of these datasets are highly consistent (Fig. S1, Fig. S2, and Fig. S3). Therefore, for simplicity, we averaged them to represent satellite-based GPP products.






### 2.5 Site-level GPP observations

We also adopted 20 EC sites from the FLUXNET2015 dataset  (Pastorello et al., 2020) with an observation period longer than 15 years to evaluate the performance of different global GPP products. These sites included 5 vegetation types: evergreen broadleaf forest (EBF), evergreen needleleaf forest (ENF), deciduous broadleaf forest (DBF), grassland (GRA), and mixed

forest (MF), all distributed over Northern Hemisphere (Table 2). The GPP variable used in this study is GPP_NT_VUT_REF. When evaluating the global gridded GPP datasets with the site observations, the bilinear interpolation method was used to interpolate the gridded data to the specific site locations.

**Table 2. FLUXNET sites used in this study. The vegetation types are: evergreen broadleaf forest (EBF), evergreen needleleaf forest**
**(ENF), deciduous broadleaf forest (DBF), grassland (GRA), and mixed forest (MF)**

| Site name | latitude | longitude | Vegetation type | Study period |
|---|---|---|---|---|
| FR-Pue | 43.74°N | 3.60°E | EBF | 2000–2014 |
| CH-Dav | 46.8°N | 9.85°E | ENF | 1997–2014 |
| DE-Tha | 50.96°N | 13.57°E | ENF | 1996–2014 |
| US-NR1 | 40.03°N | 105.55°W | ENF | 1999–2014 |
| IT-Ren | 46.59°N | 11.43°E | ENF | 1999–2013 |
| NL-Loo | 52.17°N | 5.74°E | ENF | 1996–2014 |
| RU-Fyo | 56.46°N | 32.92°E | ENF | 1998–2014 |
| FI-Hyy | 61.85°N | 24.30°E | ENF | 1996–2014 |
| CA-Man | 55.88°N | 98.48°W | ENF | 1994–2008 |
| US-UMB | 45.56°N | 84.71°W | DBF | 2000–2014 |
| US-MMS | 39.32°N | 86.41°W | DBF | 1999–2014 |
| DK-Sor | 55.49°N | 11.64° E | DBF | 2001–2014 |
| US-Ha1 | 42.54°N | 72.17°W | DBF | 1992–2012 |
| IT-Col | 41.85°N | 13.59°E | DBF | 1996–2014 |
| CA-Oas | 53.63°N | 106.20°W | DBF | 1996–2010 |
| US-Var | 38.41°N | 120.95°W | GRA | 2000–2014 |
| DK-ZaH | 74.47°N | 20.55° E | GRA | 2000–2014 |
| US-PFa | 45.95°N | 90.27°W | MF | 1996–2014 |
| BE-Bra | 51.31°N | 4.52°E | MF | 1999–2014 |
| BE-Vie | 50.31°N | 6.00°E | MF | 1997–2014 |



## 2. 6 Leaf area index

GLASS LAI version 03 was used to compare the TRENDY model ensemble LAI (S3) because it is an input parameter for GLASS and revised EC-LUE GPP. This dataset is originated from AVHRR product before 2001 and MODIS surface reflectance product (MOD09) after 2001. Biome-specific general regression neural networks were used to fuse these two datasets. Its original spatial and temporal resolutions are 0.05º × 0.05º and eight days, respectively (Xiao et al., 2016).

## 2.7 Statistical methods used

Due to the difference among temporal and spatial resolution of each product, we resampled all GPP datasets into 0.5 º × 0.5 º through the first-order conservative remapping method:

$$\bar{F}_k \ = \ \frac{1}{A_k} \int f \ dA \tag{4}$$

where $\bar{F}_k$ is the area-averaged destination quantity, $A_k$ means the area of grid k, and f is the quantity in an old grid with an overlapping area with the destination grid. We resampled NIRv and LAI into 0.5 º × 0.5 º using bilinear interpolation and generated the annual datasets according to the weights of days. We then calculated the global and regional total GPP time series from each GPP dataset and generated the linear trends of each dataset at the pixel level to generate the spatial patterns of GPP trends and at the global and zonal scales to detect the historical changes in GPP. The linear trend was calculated as

$$y \ = \ kx \ + \ b \ + \ \varepsilon, \tag{5}$$

where k represents the linear trend of the time series, b is the intercept, and $\varepsilon$ is the error term.

Finally, non-parametric Mann-Kendall trend tests were used to evaluate the level of significance for each GPP time series because it does not acquire the data to follow the normal distribution (Khaled H. Hamed, 1998).

## 3 Results and Discussions

### 3.1 Different GPP trends in DGVMs and satellite-based products

During 1982–2015, the spatial patterns of climatological annual GPP from different products are highly correlated with satellite-derived NIRv with their spatial correlation coefficients ranging from 0.84 to 0.94 (Table 3). However, the spatial distributions of the various GPP products and NIRv trends are quite different. The spatial patterns of trends of NIRv, the DGVM ensemble, FLUXCOM, and satellite-based GPP during 1982–2015 are presented in Figure 1. NIRv clearly shows increasing trends in most land regions, especially in the northwest parts of Eurasia, and shows decreasing trends over Alaska and Kazakhstan (Fig. 1a). The global distribution of the DGVM ensemble GPP trends is generally consistent with satellite-





derived NIRv with their spatial correlation coefficient ($r$) of 0.49. However, the increasing trends of the DGVM ensemble GPP in the tropical regions (Amazon and equatorial Africa) are higher than those in the boreal zone. Further, for DGVM ensemble GPP, there are about 59.8% of the global land regions showing significant positive GPP trends, and 3.8% showing significant negative GPP trends. For NIRv, 88.2% of the global land had positive GPP trends (Table 3).


Although the DGVM ensemble GPP trends are close to those of NIRv than the other GPP products used here, inconsistencies exist in spatial distribution and magnitude of GPP trends among individual model simulations. Firstly, the spatial correlations among individual models and NIRv range from 0.15 to 0.48. Secondly, the GPP simulated by the DLEM shows increasing trends in about three-fourths (77.3%) of the global land areas, while the GPP simulated by the CLASS-CTEM has increasing

trends in only about one-third (33.9%) of the land area (Table 3). Finally, in magnitude, the trends of VEGAS GPP appear generally weaker than other models (Fig. S2, Fig. S4a).

The spatial distributions of the trends between NIRv and the remaining non-DGVM products are quite different, ranging from uncorrelated to negatively correlated (Table 3). For FLUXCOM, owing to the lack of $CO_2$ fertilization effect, the GPP trend

pattern generally shows no significant trends over 74.1% of the land areas. The most striking differences between NIRv and satellite-based GPP products are located in the low latitudes, especially over Amazon and Indonesia, with the latter indicating significant decreasing trends over these regions.

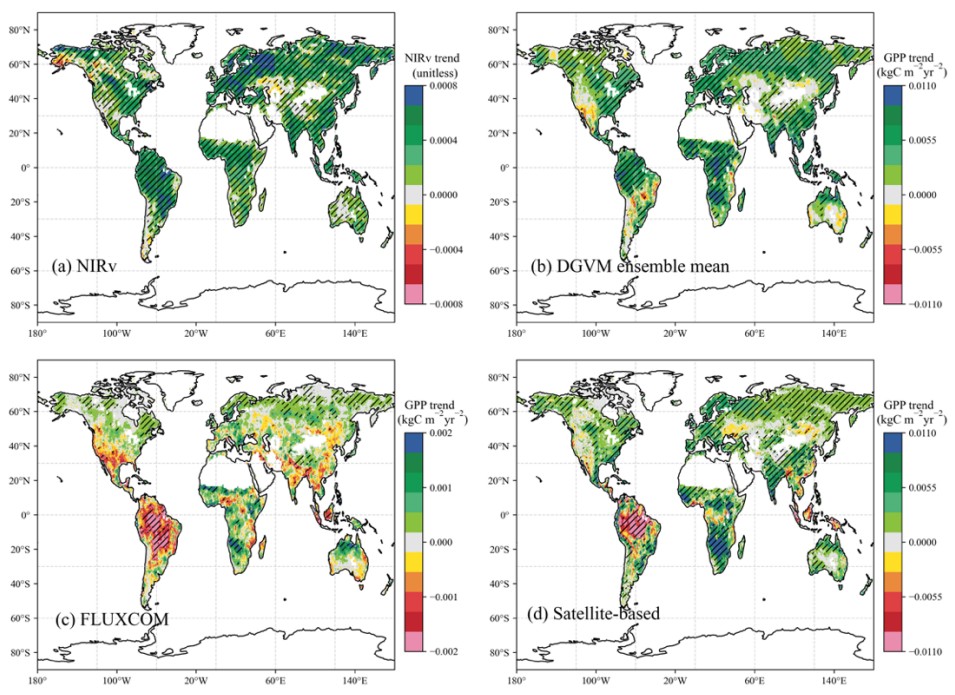



**Figure 1: Geographical distributions of linear trends of NIRv and GPP during 1982–2015. (a) AVHRR NIRv, (b) Ensemble mean of TRENDY multi-model simulated GPP, (c) FLUXCOM GPP, (d) Mean of satellite-based products from revised EC-LUE and GLASS GPP. Stripped areas indicate that the trend is significant with p < 0.05 following the non-parametric Mann-Kendall trend test. The trends of NIRv and GPP are unitless and in kgC m⁻² yr⁻², respectively. Additionally, owing to lack of the $CO_2$ fertilization effect in FLUXCOM GPP (c), we used a smaller scale than in (b) and (d).**

**Table 3. Spatial information for NIRv and different GPP products.**

| Products | Spatial correlations of climatological annual GPP with NIRv | Spatial correlations of annual GPP trends with NIRv trends | Area percentage with significant positive trends (%) | Area percentage with no significant trends (%) | Area percentage with significant negative trends (%) |
|---|---|---|---|---|---|
| TRENDY ensemble | 0.94 | 0.49 | 59.8 | 36.4 | 3.8 |
| CABLE | 0.89 | 0.39 | 56.2 | 41.1 | 2.7 |
| CLASS-CTEM | 0.90 | 0.38 | 33.9 | 61.3 | 4.8 |
| CLM4.5 | 0.85 | 0.42 | 42.6 | 55.0 | 2.4 |
| DLEM | 0.84 | 0.44 | 77.3 | 20.4 | 2.3 |
| ISAM | 0.93 | 0.42 | 64.5 | 31.9 | 3.6 |
| OCN | 0.94 | 0.48 | 59.4 | 37.8 | 2.8 |
| ORCHIDEE-MICT | 0.90 | 0.20 | 52.6 | 41.9 | 5.5 |
| ORCHIDEE | 0.89 | 0.20 | 49.3 | 45.6 | 5.1 |
| VEGAS | 0.89 | 0.15 | 51.4 | 37.4 | 11.2 |
| VISIT | 0.86 | 0.22 | 51.4 | 42.7 | 5.9 |
| GLASS | 0.95 | -0.01 | 46.4 | 42.6 | 11.0 |
| Revised EC-LUE | 0.91 | -0.03 | 48.7 | 40.7 | 10.6 |
| FLUXCOM | 0.93 | -0.26 | 15.1 | 74.4 | 10.5 |
| NIRv | - | - | 88.2 | 8.3 | 3.5 |

As mentioned above, the trends of DGVM ensemble GPP and NIRv show relatively consistent patterns for 1982–2015. However, the DGVM ensemble mean GPP shows slightly stronger trends over the tropics than over Northern Hemisphere, whereas NIRv has relatively stronger increasing trends over Northern Hemisphere (Fig. 2a). The tropical region shows the





most extensive inter-model spread for the DGVM ensemble, with the strongest trends in CABLE and the weakest trends in VEGAS (Fig. S4a, Fig. S5b). The latitudinal distribution of satellite GPP products is quite different from DGVM ensemble and NIRv. Satellite-based GPP shows a significant decrease over the tropical region, rebounding to the most substantial increase located between 15°S and 25°S. The GPP increases in the middle and high latitudes of the northern hemisphere, but

its magnitude is weaker than that of the DGVM ensemble in the most northern regions. There are sufficient numbers of pixels available in these regions to lend confidence to our results (Fig. 2c).

The GPP trends are quite different for the long-term (1982–2015) and recent short-term (2001–2015) periods (Fig. 2b, Fig. S4, Fig. S6) (Hashimoto et al., 2013; Yuan et al., 2019; Madani et al., 2020). Between 2001 and 2015, the DGVM ensemble mean

GPP trend increases more than NIRv over tropical regions and is consistent with NIRv in middle and high latitudes. However, satellite-based GPP products indicate a stronger GPP decreasing signal over the tropical region after 2001 compared to the long-term trend (Fig. 2).

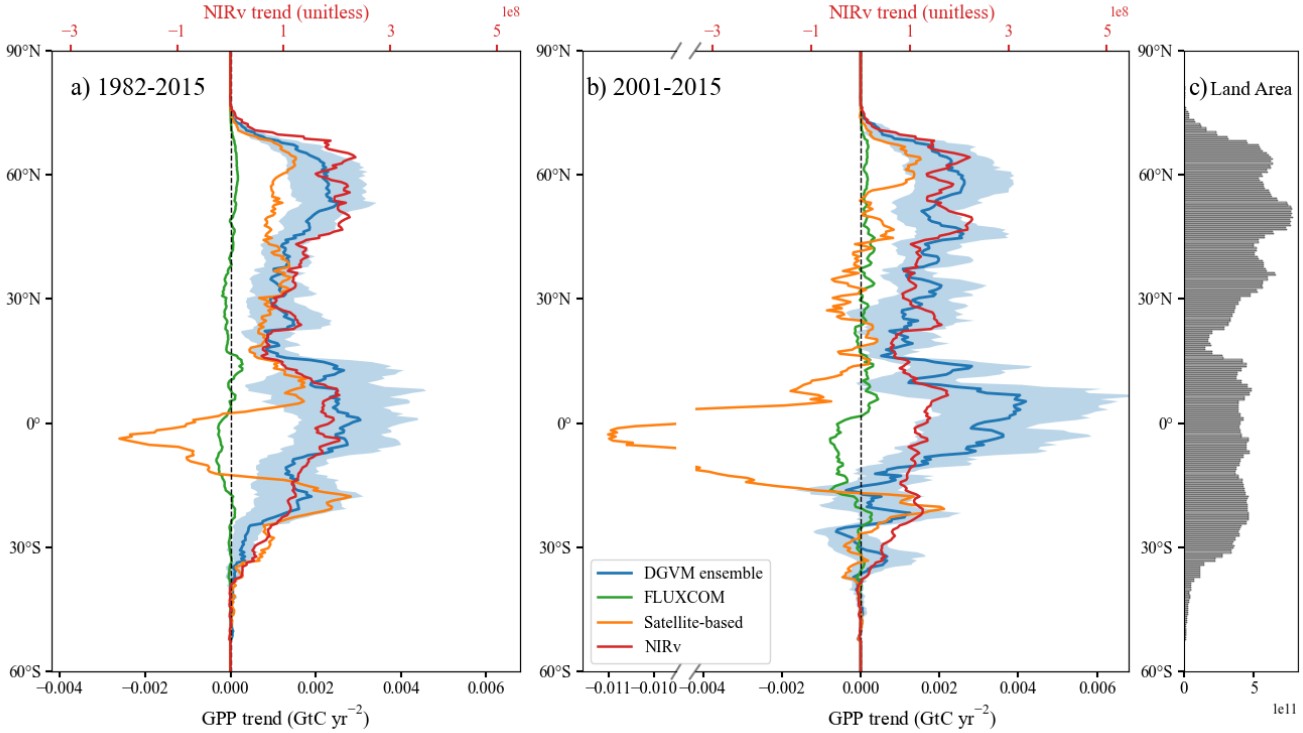

**Figure 2. Latitudinal profiles of trends of annual zonal total NIRv and GPP (0.5° latitudinal bins). Results for the DGVM ensemble mean (blue), FLUXCOM GPP (green), satellite-based products (orange), and NIRv (red) during 1982–2015 (a) and 2001–2015 (b), respectively. The GPP values are given on the bottom axis and the NIRv values on the top axis. The shaded areas represent the standard deviation of the individual TRENDY model simulated GPP trend. The units of the NIRv and GPP trends are unitless and GtC yr⁻², respectively. (c) represents the change of vegetated land area along the latitudes.**




The relative changes of annual total GPP and the linear trends of GPP among the DGVM simulations, FLUXCOM, and satellite-based GPP products, vary substantially both globally and regionally (Fig. 3, Fig. S3). Based on the analysis over the 34 years, the trend of global GPP was about 0.37 (DGVM ensemble mean), 0.0 (FLUXCOM), and 0.18 (satellite-based GPP products) GtC year$^{-2}$, respectively (Fig. 3d). Before 2001, the GPP trend of satellite-based products was generally stronger than that of DGVMs (Fig. 3a). Separating the global land into the tropics plus extra-tropical southern hemisphere (Trop+SH: 90°S–23°N) and extra-tropical northern hemisphere (NH: 23°N–90°N), results show that more than half of the increase of GPP in the DGVM ensemble is from the Trop+SH (57%). In comparison, the increase of GPP in satellite-based GPP products is mainly attributed to the NH (60%). In individual models, the majority of them show similar results to the model ensemble mean. With the exception of ORCHIDEE-MICT and VEGAS, the others indicate that Trop+SH largely contributes to the global GPP trend (ranging from 54.3% to 65.3%) (Fig. 3d).

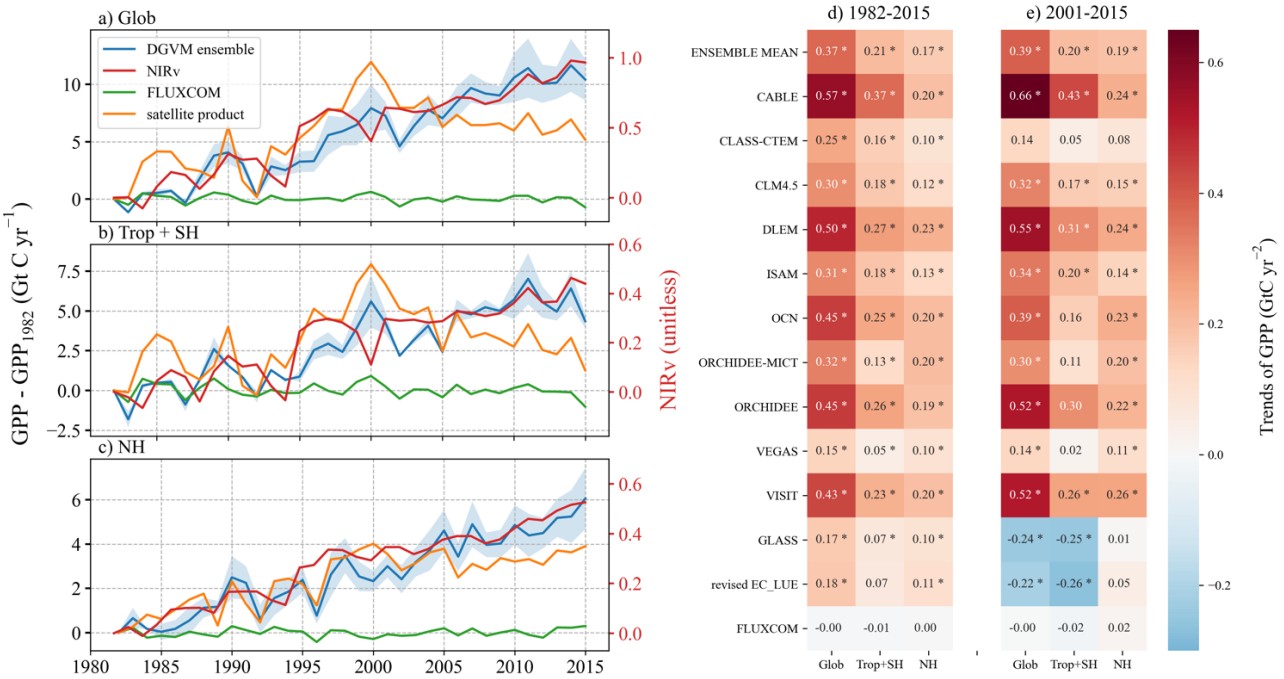

**Figure 3. Linear trends of global and regional total GPP. (a, b, and c)** Changes of annual total GPP relative to 1982, based on DGVM ensemble mean (blue), satellite-based products (orange), and FLUXCOM (green), compared to AVHRR-NIRv (red), for global (a), Trop+SH (b), and NH (c). The shaded areas denote the TRENDY inter-model spread (the standard deviation of the annual aggregated time series). **(d and e)** Global and regional GPP trends in individual models and products for the period of 1982–2015 and 2000–2015, respectively. Asterisks indicate that the trend is significant with $p < 0.05$ following the non-parametric Mann-Kendall trend test.





After 2000, there were obvious differences in the trend between DGVM ensemble and satellite-based GPP products, with the satellite-based GPP products showing an obvious turning point (Figs. 3a-c). Both GLASS and revised EC-LUE GPP changed from significant increasing trends to significant decreasing trends, resulting mainly from Trop+SH (Figs. 3a-c). Studies based on satellite-based GPP products suggested that this transition was mainly due to the increasing atmospheric vapor pressure deficit in the tropical zones (Yuan et al., 2019; Madani et al., 2020). Meanwhile, the increasing GPP trend in the NH was

greatly weakened (from 0.10 to 0.01 GtC year$^{-2}$ for GLASS and from 0.11 to 0.05 GtC year$^{-2}$ for revised EC-LUE). In contrast, DGVM ensemble mean GPP and NIRv kept increasing. However, in detail, four out of ten DGVM models (CLASS-CTEM, OCN, ORCHIDEE-MICT, and VEGAS) simulated weakened GPP increasing signals primarily from the Trop+SH (Fig. 3d and e). Similar to the spatial distribution, the FLUXCOM GPP has no noticeable trends in the study period.

**3.2 Trend attributions in DGVMs**

We analyzed the contributions of three drivers ($CO_2$ fertilization effect, climate change, and LUC) to the GPP trends during 1982–2015 by using the results of the TRENDY sensitivity experiments (Fig. 4). Globally, DGVM ensemble results suggest that the $CO_2$ fertilization effect is the dominant driver to increasing GPP (0.29 GtC year$^{-2}$ accounting for 83.9%), followed by climate change (0.09 GtC year$^{-2}$ accounting for 26.5%). Additionally, LUC has little effect on the trend of GPP (−0.04 GtC

year$^{-2}$ accounting for −10.4%). For individual model simulations, the contributions from $CO_2$ fertilization effect, climate change, and LUC range from 65.7% to 116.3%, 2% to 50.4%, and −18.4% to 5.5%, respectively. The model simulations for the period of 2001–2015 show similar results (Fig. S7).

The spatial distributions of GPP trends indicate that the $CO_2$ fertilization effect consistently increases global GPP, especially

in tropical rainforest areas (Fig. 5a). Climate change has inhomogeneous effects on GPP owing to different regional changes of climate elements (Fig. 5c and Figs. S8a and b) associated with different vegetation sensitivities to each climate element (Wang et al., 2016; Jung et al., 2017). For instance, in the high northern latitudes, global warming dominates the increase of GPP (Fig. 5c, Figs. S8a, and S9c); the increase in temperature and decrease in precipitation over Amazon lead the GPP to decrease; the increase of GPP over Equatorial Africa appears to be consistent with the increase of the precipitation based on

visual comparison against the trends for the CRU observational dataset (Fig. 5c and Fig. S8b). Concentrated in the Trop+SH, LUC basically weakens GPP (Fig. 5e and Fig. S9a), mainly due to deforestation (Friedlingstein et al., 2019).



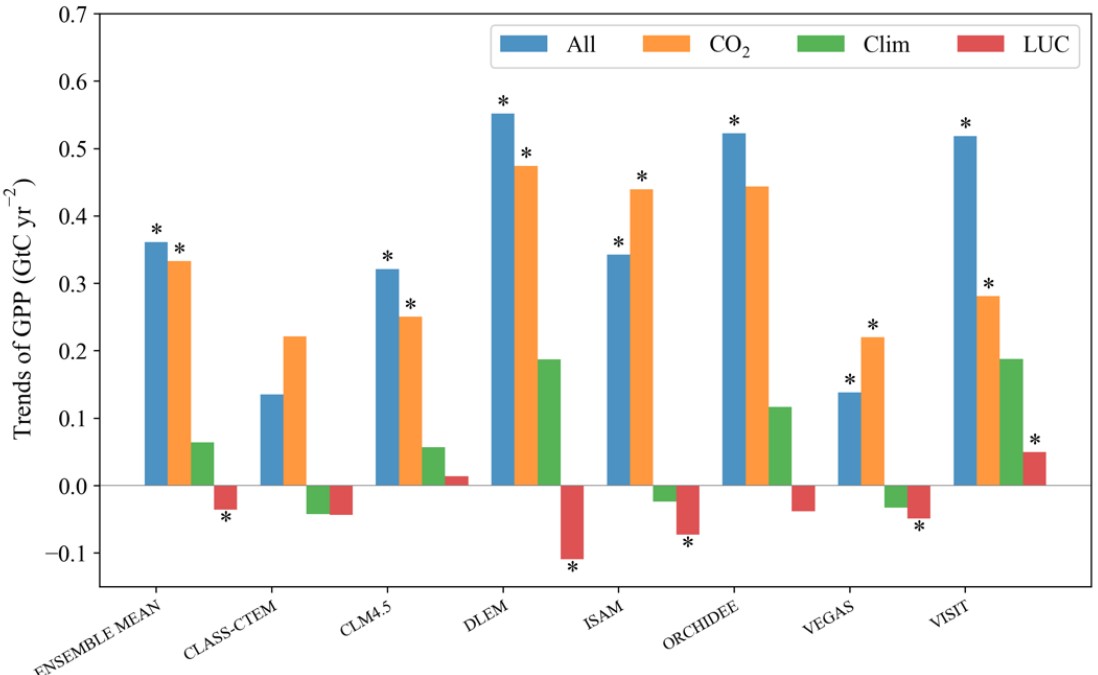

**Figure 4. Attributions of global total GPP trends for TRENDYv6 simulations: $CO_2$ fertilization effect (S1), climate (S2-S1), and land-use change (S3-S2). "All" gives the values of the reference simulation that includes the effect of all three drivers (S3). Asterisks indicate that the trend is significant with $p < 0.05$ following the non-parametric Mann-Kendall trend test.**

## 3.3 Uncertainties in GPP trends

### 3.3.1 DGVM simulations

By analyzing the simulation results driven by each factor, it can be found that though the $CO_2$ fertilization effect has the most considerable contribution to the global GPP trend, it has the largest inter-model uncertainty ($\sigma = 0.11$ GtC yr$^{-2}$) among three drivers (Fig. 4). A previous study showed that some models might overestimate the $CO_2$ fertilization effect on stomatal closure (Anav et al., 2015). The spatial pattern of the standard deviation among each model indicates that the inter-model spreads are mainly located in the Trop+SH (Fig. 5h). The inter-model spread attributed to the $CO_2$ fertilization effect shows a consistently positive effect on GPP at the global scale (Fig. 5a, b). By contrast, the inter-model spreads driven by climate change and LUC over Trop+SH outweigh the $CO_2$ fertilization effect (Figs. 5b, d, and f). However, because the inhomogeneous impacts on GPP from climatic elements can offset each other to a large extent (Fig. 5c), it makes the largest inter-model uncertainty of $CO_2$ fertilization to the global GPP increase rather than the climate effect. Meanwhile, the largest uncertainty in the impact of LUC on GPP increase concentrates over 20º–40ºS in South America.





We further calculated the spatial correlation coefficients among the GPP trend of each model simulation to quantify their spatial consistencies. The correlation coefficient between each model varied from 0.16 to 0.61 (Fig. S10), implying that large uncertainties existed in the distribution of GPP trends among models, which were caused by differences in model structures and parameterizations (Rogers, 2014; Rogers et al., 2017). Furthermore, studies have shown that the global GPP increase will

be largely overestimated without nitrogen (N) constraints, especially in the tropical region, where the nitrogen limitation will reduce the photosynthetic capacity of vegetations and weaken its response to the increasing atmospheric $CO_2$ concentration (He et al., 2017; Terrer et al., 2019). Phosphorus (P) availability also limits the extent to which plants respond to the $CO_2$ fertilization effect, which is especially relevant in the Amazon forest (Fleischer et al., 2019). Therefore, DGVM ensemble GPP may overestimate the increasing GPP trend in the tropical regions since not all of the models used in this study take the effect

of N limitation into consideration, and no model includes P limitation to the $CO_2$ fertilization effect.

**Figure 5. The spatial distribution patterns and the inter-model spreads of GPP trends from the DGVM ensemble. (a and b) GPP trends and spreads owing to CO₂ fertilization effect; (c and d) GPP trends and spreads owing to climate change; (e and f) GPP trends and spreads owing to LUC; (g and h) GPP trends and spreads from the combined effects of all drivers (S3). Stripped areas indicate that the trend is significant with p < 0.05 following the non-parametric Mann-Kendall trend test.**

### 3.3.2 Satellite-based GPP products

In this study, GLASS GPP and revised EC-LUE GPP were used as a representative of long-term satellite-based GPP products.

As an essential input in the LUE model (Eq. 1), APAR is a function of LAI, suggesting that LAI is a key parameter in satellite-





derived GPP. The spatial correlation coefficient of trends between GLASS LAI and satellite-based GPP (i.e., the mean of
      GLASS GPP and revised EC-LUE GPP) is 0.42. A previous study over China has shown that satellite-based LAI datasets play
      a more important role in GPP estimation than meteorological data for all land cover types (Liu et al., 2014). Also, the spatial
      distribution of trends between LAI and GPP simulated from the DGVM ensemble is even more consistent ($r$ = 0.77),
      confirming the previous studies showing that the trends of GPP and LAI are highly correlated in biome models (Ito et al., 2017;
Liu et al., 2019), and the changes of GPP and LAI are consistent in earth system models from CMIP5 (Hashimoto et al., 2019).

      Figure 6 compares the LAI trends of GLASS and the DGVM ensemble during 2001–2015. The spatial distribution of DGVM
      LAI indicates significant increasing trends over the boreal forest region, Indonesia, Equatorial Africa, and India, and significant
      decreasing trends over Kazakhstan, Southeast Asia, and Western Australia. The trends of GLASS LAI were obviously weaker
than that of the DGVM ensemble, especially in the equatorial Africa region, northern Amazon region, Indonesia, and Northern
      high latitudes. The large inconsistencies between the LAI from the DGVMs and those observed in the satellite products could
      lead to substantial uncertainties in generating/simulating the global GPP (Fig. 1 and Fig. 6). Xiao et al., (2017) suggested that
      the trends of four satellite-derived LAI products showed large discrepancies in equatorial Africa from 1982 to 2011 and
      differed across each vegetation type. Jiang et al., (2017) revealed that NOAA satellite orbit changes and MODIS sensor
degradation might cause long-term satellite-derived LAI products inconsistent with each other. Xie et al., (2019) also suggested
      that satellite-derived LAI datasets can cause uncertainties in GPP estimations through model structure and the complexity of
      the terrain. Hence, the long-term trends of satellite GPP products based on satellite-derived LAI remain highly uncertain (Smith
      et al., 2016; Jiang et al., 2017; Liu et al., 2018).



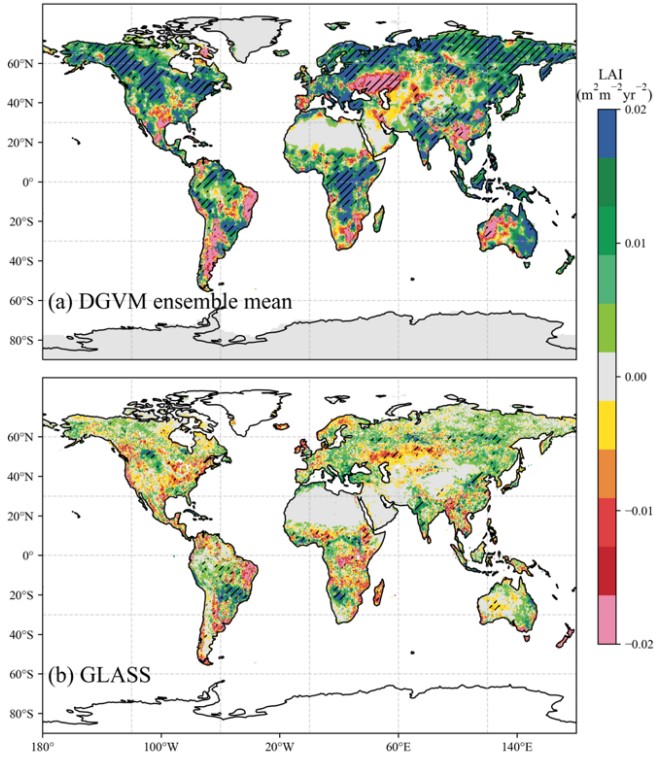

**Figure 6. The spatial distributions of LAI trend (m² m⁻² yr⁻²) from (a) DGVM ensemble mean and (b) GLASS from 2001 to 2015. Striped areas indicate that the trend is significant with p < 0.05 following the non-parametric Mann-Kendall trend test.**

### 3.3.3 Evaluation of site-level GPP trends

We further adopted 20 sites with observations longer than 15 years from FLUXNET2015 datasets to evaluate long-term GPP trends from global products and simulations. Compared to the site observations, the magnitudes of GPP at most of site locations were underestimated by satellite-based GPP products, FLUXCOM, and the DGVM ensemble mean. Also, the interannual variation and long-term trend of GPP at sites are more obvious than those of global GPP products and NIRv (Fig. 7). This is possibly due to the fact that climate variables at 0.5-degree grid cells are smoothed out compared to those recorded at individual

sites, which leads to a moderate GPP variation.

More than half of the sites indicate that FLUXNET GPP has increased on a long-time scale (Fig. 7, Fig. S11), which was mainly caused by rising LUE due to the $CO_2$ fertilization effect and increased green vegetation cover (Cai and Prentice, 2020). Although FLUXCOM was upscaled from FLUXNET datasets, it did not capture the trends of GPP observed at sites, which

was also mentioned by Anav et al. (2015). Sites with significant increasing GPP trends were all captured by DGVM ensemble





mean, but some were missed by satellite-based GPP (Figs. 7b, d, k, l, m, s, and t). Furthermore, none of sites with decreasing GPP trends were reflected in the global GPP products and NIRv (Figs. 7f, h, i, j, n, q, r), which may be in part due to the different spatial representativenesses between a tower fetch and a model or satellite grid point. Therefore, the uncertainty remained when using the site-level observed GPP to evaluate the GPP trends of the DGVM simulations and satellite-based

GPP products. It is worth mentioning that sites with more than 15 years of observations were all located at NH (from 39.32°N to 50.96°N) (Table 2). It is hardly for us to evaluate the GPP trends of global products over the Trop+SH by using the GPP observations from sites.





**Figure 7. Comparisons of annual GPP over different FLUXNET2015 sites (black), FLUXCOM (green), satellite-based product (orange), DGVM ensemble (blue), and NIRv (red). The global GPP datasets were interpolated into the locations of these 20 sites according to the bilinear interpolation method. Observation sites with significant trends are marked with values. Single (\*) and double (\*\*) asterisks indicate that the trend is significant with p < 0.1 and p < 0.05 following the non-parametric Mann-Kendall trend test. The units of GPP and GPP trend are kgC m$^{-2}$ yr$^{-1}$ and kgC m$^{-2}$ yr$^{-2}$ respectively.**


## 4 Conclusions

Based on five kinds of GPP or GPP-related datasets, including satellite-based products, machine learning models, DGVM simulations, satellite-observed proxy (NIRv), and site-level observations, we comprehensively assessed the global and regional GPP trends during 1982−2015. The simulated spatial pattern of GPP trends from the DGVM ensemble is highly consistent

with NIRv, but shows considerable inconsistency with satellite-based GPP products, especially in the tropical regions. After 2000, the GPP generated by the satellite-based GPP products decreased significantly in Trop+SH, and the increasing trend in NH also weakened. However, the results of DGVMs showed that global GPP kept increasing after 2000, even in the tropical region, which was closer to the performance of NIRv. By analyzing the impact of each driving factor in DGVM simulations, the results indicate that the $CO_2$ fertilization effect has the dominant contribution to the global GPP. Spatially, the $CO_2$

fertilization effect makes the global GPP increased consistently, while climate has inhomogenous impact on GPP trends over different regions.

We further explored the uncertainties in GPP trends among these different datasets. For DGVM ensemble, globally, the $CO_2$ fertilization effect causes the largest inter-model spread. At the grid cell level, the uncertainties in simulated GPP trends

concentrate over the Trop+SH, which result mainly from climate and LUC. Furthermore, the large discrepancy in the GPP trends between DGVM ensemble and satellite-based GPP products is, to a large extent, induced by the difference of vegetation canopy structure parameter (LAI). Therefore, the highly uncertain satellite-derived LAI data in the tropical regions increase the uncertainty of satellite GPP products and weaken their reliability in explaining changes in the global GPP.

Finally, GPP trends from satellite-based products and DGVM simulations were evaluated by using the FLUXNET2015 dataset. Results show that all of sites with significant increasing GPP trends can be captured by DGVM ensemble mean, but some of them were missed by satellite-based GPP. And none of sites with decreasing GPP trends were reflected in the global GPP products. Therefore, uncertainty remained when using the FLUXNET observed GPP to evaluate the GPP trends of the global GPP products.


Generally, the differences among models, observations, and products suggest the importance of the research on the GPP trend and make our caution to interpret the mechanisms of the global carbon cycle by using the long-term GPP products.



**Data Availability.** All data acquired or used in this analysis are available from the links in Table S1.


**Author contributions.** ZN, WJ, and YRQ conceived and designed this study. YRQ and TWH completed the statistical analysis and prepared figures. YRQ and WJ drafted the manuscript with contributions from all authors.

**Competing interests.** The authors declare that they have no conflict of interest.


**Acknowledgments.** We gratefully acknowledge the TRENDY DGVM community and all people and institutions who provided datasets used in this study. We thank discussion with Shengjie Zhou, Huihang Sun, Xiaohui Lin, Xing Yan, Han Mei, and Zhiqiang Liu. This work was supported by the National Key R&D Program of China (No. 2017YFB0504000). Jun Wang was supported by the National Natural Science Foundation of China (41807434), National Key R&D Program of China
(2020YFA0607504), and Fundamental Research Funds for the Central Universities (0904-14380028). MJM has been supported by the H2020 European Research Council (grant no. 776810).

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
