# Peer review of "Divergent historical GPP trends among state-of-the-art multi-model simulations and satellite-based products"

_Earth System Dynamics, 2021_

## Referee Comment (RC2)

Review report of manuscript esd-2021-69 "Divergent historical GPP trends among state-of-the-art multi-model simulations and satellite based products"

This study investigates global and regional GPP trends during 1982-2015, based on Dynamic Global Vegetation Model (DGVM) TRENDY v6 multi-model simulations, machine learning technique based GPP products, multiple GPP data sets derived from satellite-based products, FLUXNET observed GPP products, Long-term satellite-based near-infrared radiance of vegetation (NIRv) products. DGVM ensemble is consistent with NIRv but inconsistent with satellite-based GPP products and FLUXNET observations. Significant uncertainties existed in the distribution of GPP trends among models. Most of the models used in this study have not considered Nitrogen limitation and Phosphorous availability. The manuscript re-visits all the above products and discusses the limitations. The manuscript deals with important datasets, but the results are not easy to connect.

Review Comments:

(MAJOR REVISION)

1) Abstract: "Machine Learning Technique"…….study uses the FLUXCOM product. No machine learning algorithm was used, so this sentence may be revised.

2) Abstract: "Trends after 2000 was different from the full time-series….", Satellite-based GPP product showed a decreasing trend, and DGVM showed an increasing trend mainly because LAI was not represented well in satellite-product. Was LAI represented well in TRENDY model simulations? Also, why trends after 2000 was different from full-time series?

3) Section 2.2: FLUXCOM GPP product area percentage with no significant trend is 74.4% (Table 3), highest among all products. The spatial correlation of the annual GPP trend with NIRv is -0.26. However, the spatial correlation of climatological annual GPP trend with NIRv is +0.93. Can you explain this contrasting feature? Also, revise this section for clarity.

4) Section 2.3, last line: "….datasets are highly consistent…..", Spatial pattern and temporal changes of these datasets are NOT highly consistent. There are

differences, e.g, ORCHIDEE shows a much stronger signal than VEGAS over South America and Africa. Revise this section.

5) Section 2.5: FLUXNET observations used in this study are distributed over Northern Hemisphere only. However, the performance of GPP products evaluated with these observations is distributed globally. How are the Southern Hemisphere GPP products evaluated?

6) Section 2.6: Message not clear. Describe the section in detail.

6) Section 2.7, line 180: "We then calculated the global and regional……historical changes in GPP". Message not clear; revise the sentence

7) Linear trend calculated. Whether all the GPP datasets in each pixel show a linear trend?

8) NIRv should be discussed as a separate sub-section in the Datasets and Methods section

9) Fig1(a): NIRv is a unitless variable. Is it an index like NDVI or EVI? The magnitude of NIRv is in the order of the fourth decimal. Is it possible to normalize it between 0 to 1 for better representativity?

10) Fig.1(b): "DGVM ensemble mean". However In Table-3, it is represented as "TRENDY ensemble". Be consistent with naming.

11) Section 3.1, line 235: "…..consistent with NIRv in middle and high latitudes" is not valid for Southern Hemisphere.

12) Fig 2 (a,b): FLUXCOM dataset GPP trend is near zero in most latitude bands. Describe this in the text.

13) NIRv trend is in the fourth decimal number in Fig.1 (a). However, it varies between 1 to 3 in Fig.2(a). Why such discrepancy?

14) Fig.2(a,b): DGVM ensemble and NIRv products are higher in latitude bands -20:+20. However, the change of vegetated land area is greater in +30:+60 latitude bands. Why?

15) Line 250: "In comparison, the increase of GPP in satellite-based GPP products …………to the NH(60%)". Valid for Trop+SH as well. Revise it accordingly.

16) Fig.3 (a,b): The highest trend has shown during the year 2000. However, it is absent in the Northern Hemisphere (NH).

---

## Author Comment (AC1)

**Correction figure 4 for "Divergent historical GPP trends among state-of-the-art multi-model simulations and satellite-based products"**

Ruqi Yang et al.,

5    The fourth figure in the manuscript (Fig. 4) calculated the global GPP trend from 2001 to 2015 and should be placed in the supplement (Fig. S7), but I put it in the article by mistake. Figure 4 should be the GPP trend from 1982-2015, which is presented here.

[Figure]

**Figure 4. Attributions of global total GPP trends for TRENDYv6 simulations from 1982 to 2015: $CO_2$ fertilization effect (S1), climate**
10   **(S2-S1), and land-use change (S3-S2). "All" gives the values of the reference simulation that includes the effect of all three drivers (S3). Asterisks indicate that the trend is significant with $p < 0.05$ following the non-parametric Mann-Kendall trend test.**

---

## Author Comment (AC2)

Reply to reviewer's comment from Anonymous Referee #1:

**Dear Editor and Reviewer,**

   We want to thank the reviewer for the valuable comments, which helped improve our manuscript. We address all of the reviewer's comments below and describe how the suggested changes have been implemented in the revised version of the manuscript.

**Ruqi Yang et al. shows spatial patterns of GPP trends based on multiple datasets including NIRv, satellite-based, and DGVM and found out differences between datasets. I believe this information is useful to study global GPP changes and the uncertainty of each dataset. I have only a few minor comments on this manuscript.**

**L79 While NIRv is introduced in the abstract, it is needed to describe here again. "Long-term satellite-based near-infrared radiance of vegetation (NIRv)"**

Reply: Thank you for your careful review. We have revised accordingly.

**L90 Instead of mentioning TRHENDYv6 here, how about simulations from process-based models?**

Reply: Thanks for your suggestion. We have modified the "TRENDYv6" to "Process-based models" in the revised manuscript.

**L96 I think the explanation of "NIRv" is missing in section 2 (Datasets and methods). I believe the authors need to explain details about NIRv in section 2 (should be 2.1). For example, how this dataset was gained and evaluated in the previous literature.**

Reply: Thanks for your comment. We have added the NIRv descriptions in section 2.4.

**L248 Can you provide 95% confidence intervals for the trend of global GPP such as 0.37 ± ??? (DGVM ensemble mean)**

Reply: Thanks for your suggestion. We have modified the sentence as "the trend of global GPP was about 0.37 ± 0.08 (DGVM ensemble mean ± 95% confidence intervals)".

**L403 And -> Also,**

Reply: Thanks for your comment. We have modified the "And" to "Also" in the revised manuscript.

---

## Author Comment (AC3)

Reply to reviewer's comment from Anonymous Referee #2:

**Dear Editor and Reviewer,**

We genuinely thank the reviewer for his suggestions and comments which helped improve our manuscript. We tried our best to address all of the reviewer's comments point by point and described how the relative modifications have been made in the revised version of our manuscript.

**This study investigates global and regional GPP trends during 1982-2015, based on Dynamic Global Vegetation Model (DGVM) TRENDY v6 multi-model simulations, machine learning technique based GPP products, multiple GPP data sets derived from satellite-based products, FLUXNET observed GPP products, Long-term satellite-based near-infrared radiance of vegetation (NIRv) products. DGVM ensemble is consistent with NIRv but inconsistent with satellite-based GPP products and FLUXNET observations. Significant uncertainties existed in the distribution of GPP trends among models. Most of the models used in this study have not considered Nitrogen limitation and Phosphorous availability. The manuscript re-visits all the above products and discusses the limitations. The manuscript deals with important datasets, but the results are not easy to connect.**
**1) Abstract: "Machine Learning Technique" ……. study uses the FLUXCOM product. No machine learning algorithm was used, so this sentence may be revised.**

Reply: Thank you for your comment. We have revised the sentence as "Using long-term satellite-based near-infrared radiance of vegetation (NIRv), a proxy for GPP, and multiple GPP datasets derived from satellite-based products, Dynamic Global Vegetation Model (DGVM) simulations, and an upscaled product from eddy covariance (EC) measurements, here we comprehensively investigated their trends and analyzed the causes for any discrepancies during 1982–2015."

**2) Abstract: "Trends after 2000 was different from the full time-series….", Satellite-based GPP product showed a decreasing trend, and DGVM showed an increasing trend mainly because LAI was not represented well in satellite-product. Was LAI represented well in TRENDY model simulations? Also, why trends after 2000 was different from full-time series?**

Reply: Thanks for your comment.
a) Previous studies based on satellite-based products showed a turning point of GPP in 2000 and suggested that it was caused by the increased atmospheric vapor pressure deficit in the tropical zones (Yuan et al., 2019; Madani et al., 2020). Therefore, except the GPP trend analyses during the long-term period of 1982–2015, we additionally analyzed their GPP trend performance in the short-term period of 2001–2015. And we find out that satellite-based GPP product showed a decreasing trend, and DGVM kept an increasing trend. (See text section 3.1 and Fig. 3)

b) We calculated the spatial correlation coefficients of trends between GPP and LAI and found that they have high correlation coefficients (satellite: r=0.42, and DGVM ensemble: r=0.77), indicating that LAI is a key parameter accounting for GPP trend behaviors in this study. Therefore, in section 3.3.2, we compared satellite-derived LAI which was used to generate satellite-derived GPP products and DGVM simulated LAI, in order to explain the different GPP trends after 2000 between satellite and DGVM GPP.

c) Further, we pointed out that satellite-derived LAI product has large uncertainties themselves (Xiao et al., 2017; Jiang et al., 2017), cascading to influence the satellite-derived GPP products (Xie et al., 2019) (section 3.3.2). So we suggest that the uncertainty in satellite-based GPP products induced by highly uncertain LAI data in the tropics undermines their roles in assessing the performance of DGVM simulations. It is worth mentioning that we cannot judge whether the LAI was represented well in DGVMs because of the lack of reliable large-scale LAI observations. In the text, we also used a new GPP proxy, NIRv, and pointed out that DGVM GPP trends closely resemble NIRv trends, proving that the trends from DGVM may have better performance than satellite-derived GPP products..

**3) Section 2.2: FLUXCOM GPP product area percentage with no significant trend is 74.4% (Table 3), highest among all products. The spatial correlation of the annual GPP trend with NIRv is -0.26. However, the spatial correlation of climatological annual GPP trend with NIRv is +0.93. Can you explain this contrasting feature? Also, revise this section for clarity.**

Reply: Thank you for your comments. That is correct. The spatial correlation of the climatological GPP between NIRv and FLUXCOM GPP is 0.93, and the spatial correlation of the annual GPP trend between NIRv and FLUXCOM GPP is -0.26. This negative correlation is due to the lack of $CO_2$ fertilization effect in FLUXCOM (Jung et al., 2020), so this dataset has a very weak trend (Fig. 1, Fig. 2, Fig. 3). Our result confirms the author's assessment and is also mentioned by (Anav et al., 2015).

We have clarified the sentence in section 2.2 as "*The FLUXCOM datasets comprised of 120 global carbon flux products generated by nine machine learning techniques based on site-level observed GPP measured by EC associated with remote sensing information and meteorology data, but did not take the $CO_2$ fertilization effect into account (Jung et al., 2020).*"

**4) Section 2.3, last line: "….datasets are highly consistent…..", Spatial pattern and temporal changes of these datasets are NOT highly consistent. There are differences, e.g, ORCHIDEE shows a much stronger signal than VEGAS over South America and Africa. Revise this section.**

Reply: Thanks for your comments. "The high consistency" here refers to the two satellite-derived products, the GLASS GPP and Revised EC-LUE GPP. We have revised the sentence as "*The spatial pattern and temporal changes of the GLASS GPP and Revised EC-LUE GPP are highly consistent (Fig. S1, Fig. S2, and Fig. S3).*

*Therefore, for simplicity, we averaged them to represent satellite-based GPP products*."

**5) Section 2.5: FLUXNET observations used in this study are distributed over Northern Hemisphere only. However, the performance of GPP products evaluated with these observations is distributed globally. How are the Southern Hemisphere GPP products evaluated?**
Reply: Thanks for your comments. The study only chose the site with an observation time period longer than 15 years to evaluate the long-term GPP trends. No site located in Southern Hemisphere meets this criterion.

We also adopted 21 EC sites with an observation length of over 12 years, three of them located in Trop+SH (Table S2). We found that the results didn't change our conclusion. We have added the figures and table accordingly in the supplementary materials.

**Table S2: additional FLUXNET sites used.**

| Site name | latitude | longitude | Vegetation type | Study period |
|-----------|----------|-----------|-----------------|--------------|
| IT-Lav | 45.96°N | 11.28°E | ENF | 2003–2014 |
| FI-Sod | 67.36°N | 26.64°E | ENF | 2001-2014 |
| FR-LBr | 44.71°N | 0.77°W | ENF | 1996-2008 |
| US-Me2 | 44.45°N | 121.56°W | ENF | 2002-2014 |
| CA-TP3 | 42.71°N | 80.35°W | ENF | 2003-2014 |
| CA-TP1 | 42.66°N | 80.56°W | ENF | 2003-2014 |
| CA-TP4 | 42.71°N | 80.36°W | ENF | 2002-2014 |
| CA-Obs | 53.99°N | 105.12°W | ENF | 1999-2010 |
| IT-SRo | 43.73°N | 10.28°E | ENF | 1999-2012 |
| DE-Geb | 51.10°N | 10.91°E | CRO | 2001-2014 |
| US-Ne3 | 41.18°N | 96.44°W | CRO | 2001-2013 |
| US-Ne1 | 41.17°N | 96.48°W | CRO | 2001-2013 |
| US-Ne2 | 41.16°N | 96.47°W | CRO | 2001-2013 |
| DE-Hai | 51.08°N | 10.45°E | DBF | 2000-2012 |
| US-WCr | 45.81°N | 90.08°W | DBF | 1999-2014 |
| ZA-Kru | 25.02°S | 31.50°E | SAV | 2000-2013 |
| RU-Sam | 72.37°N | 126.50°E | GRA | 2002-2014 |
| CA-Gro | 48.22°N | 82.16°W | MF | 2003-2014 |
| AU-Tum | 35.66°S | 148.15°E | EBF | 2001-2014 |
| AU-How | 12.49°S | 131.15°E | WSA | 2001-2014 |
| US-Ton | 38.43°N | 120.97°W | WSA | 2001-2014 |

**Figure S12. Comparisons of annual GPP over different FLUXNET2015 sites (black), FLUXCOM (green), satellite-based product (orange), DGVM ensemble mean (blue), and NIRv (red) during 1996-2014. The global GPP datasets were interpolated into the locations of these 21 sites according to the bilinear interpolation method.**

[Figure]

**6) Section 2.6: Message not clear. Describe the section in detail.**

Reply: Thanks for your comment. We have described this product in more details as *"GLASS LAI version 05 was used to compare the TRENDY model ensemble LAI (S3) because it is an input parameter for GLASS GPP and revised EC-LUE GPP. This dataset is originated from version 4 Long-Term Data Record (LTDR) AVHRR surface reflectance product before 2001 with a spatial resolution of 0.05° × 0.05° and MODIS surface reflectance product (MOD09) after 2001 with a spatial resolution of 1km × 1km. The spatial-average method was used to aggregate the dataset to 0.05° × 0.05°. Biome-specific general regression neural networks were used to fuse these two datasets to generate a long-term LAI product (1982 - 2018), which improved performance than the original datasets. Its spatial and temporal resolutions are 0.05° × 0.05° and eight days, respectively (Xiao et al., 2016). The previous study have shown that this product performed well than other long time LAI estimation based on the evaluation of high-resolution reference maps at VAlidation of Land European Remote sensing Instruments sites (Xiao et al., 2017)."*

**6) Section 2.7, line 180: "We then calculated the global and regional……historical changes in GPP". Message not clear; revise the sentence**

Reply: Thank you for your comments. The sentence is modified as "*To detect the historical changes in GPP in each dataset; we calculated the global and regional total GPP and their linear trends. We also calculated the linear trends of each dataset at the pixel level to generate the spatial patterns of GPP trends*."

**7) Linear trend calculated. Whether all the GPP datasets in each pixel show a linear trend?**
Reply: Thank you for your comments. Yes, we calculate GPP trends in each pixel from all GPP datasets and show their spatial patterns in Figure S2.

**8) NIRv should be discussed as a separate sub-section in the Datasets and Methods section.**
Reply: Thank you for your careful review. We have added this information in section 2.4.

**9) Fig1(a): NIRv is a unitless variable. Is it an index like NDVI or EVI? The magnitude of NIRv is in the order of the fourth decimal. Is it possible to normalize it between 0 to 1 for better representativity?**
Reply: Thank you for your comments. NIRv is an index like NDVI and EVI and calculated as a function of monthly NDVI and near-infrared reflection of the total pixel (NIRT) via the equation of NIRv = (NDVI - 0.08) × NIRT (Badgley et al. 2017). We can normalize NIRv between 0 to 1, but the spatial pattern will not change, and Figure 2 calculated the zonal total trend of NIRv to compare GPP, so we think it is okay to keep the original value.

**10) Fig.1(b): "DGVM ensemble mean". However In Table-3, it is represented as "TRENDY ensemble". Be consistent with naming.**
Reply: Thank you for your careful review. We have modified "TRENDY ensemble" to "DGVM ensemble mean" in the revised manuscript.

**11) Section 3.1, line 235: "…..consistent with NIRv in middle and high latitudes" is not valid for Southern Hemisphere.**
Reply: Thank you for your careful review. In the original manuscript, we did not express our meaning correctly. After modification, we have rewritten the sentences as "*Comparing to the comparable trend magnitudes of DGVM ensemble mean GPP and NIRv over NH during these two periods, the DGVM ensemble mean GPP trends show a much stronger increase, but NIRv appears a little weakened increase over tropical regions during 2001–2015 (Figs. 2a and b).*"

**12) Fig 2 (a,b): FLUXCOM dataset GPP trend is near zero in most latitude bands. Describe this in the text.**
Reply: Thanks for your comment. We have added this sentence in the text as "*Also, FLUXCOM GPP trends are near zero in most latitudinal bands during these two periods (Figs. 2a and b), owing to the lack of CO2 fertilization effect (Jung et al., 2020).*"

**13) NIRv trend is in the fourth decimal number in Fig.1 (a). However, it varies between 1 to 3 in Fig.2(a). Why such discrepancy?**

Reply: Thanks for your question. Figure 2 shows the zonal total trend of NIRv. The value becomes much larger because it is multiplied by the grid area. We have added the unit of figure 2 accordingly.

**14) Fig.2(a,b): DGVM ensemble and NIRv products are higher in latitude bands 20:+20. However, the change of vegetated land area is greater in +30:+60 latitude bands. Why?**

Reply: Thanks for your comments. Figure 2c represents the latitudinal total vegetated land areas. We have rewritten the figure caption for Fig. 2c to avoid misleading readers as *"(c) represents the latitudinal total vegetated land areas"*.

**15) Line 250: "In comparison, the increase of GPP in satellite-based GPP products …………to the NH(60%)". Valid for Trop+SH as well. Revise it accordingly.**

Reply: Thank you for your comment. We have revised the sentence as *"In comparison, the increase of GPP in satellite-based GPP products is mainly attributed to the NH (60%) rather than Trop+SH (40%) (Fig. 3d)."*

**16) Fig.3 (a,b): The highest trend has shown during the year 2000. However, it is absent in the Northern Hemisphere (NH).**

Reply: Thanks for your comment. I think there might be some misunderstanding here since figure 3a, b, and c shows the relative change of GPP ($GPP_i$) to GPP in 1982 ($GPP_{1982}$) instead of the GPP trend. Satellite-based GPP products show a positive trend before 2000 and a negative trend after 2000, maybe caused by the increased atmospheric vapor pressure deficit in the tropical zones (Yuan et al., 2019). Figure 7 shows that the LAI used to generate satellite-based GPP decreases in the Trop+SH and have weaker trend in NH compared to DGVM ensemble LAI. Therefore, the transition seems slight in NH.

References:

Anav, A., Friedlingstein, P., Beer, C., Ciais, P., Harper, A., Jones, C., Murray-Tortarolo, G., Papale, D., Parazoo, N. C., Peylin, P., Piao, S., Sitch, S., Viovy, N., Wiltshire, A., and Zhao, M.: Spatiotemporal patterns of terrestrial gross primary production: A review, Reviews of Geophysics, 53, 785-818, 10.1002/2015rg000483, 2015.

Badgley, G., Field, C. B., and Berry, J. A.: Canopy near-infrared reflectance and terrestrial photosynthesis, Science Advances, 3, 10.1126/sciadv.1602244, 2017.

Jiang, C., Ryu, Y., Fang, H., Myneni, R., Claverie, M., and Zhu, Z.: Inconsistencies of interannual variability and trends in long-term satellite leaf area index products, Global Change Biology, 23, 4133-4146, 10.1111/gcb.13787, 2017.

Jung, M., Schwalm, C., Migliavacca, M., Walther, S., Camps-Valls, G., Koirala, S., Anthoni, P., Besnard, S., Bodesheim, P., Carvalhais, N., Chevallier, F., Gans, F., Goll, D. S., Haverd, V., Köhler, P., Ichii, K., Jain, A. K., Liu, J., Lombardozzi, D., Nabel, J. E. M. S., Nelson, J. A., O'Sullivan, M., Pallandt, M., Papale, D., Peters, W., Pongratz, J., Rödenbeck, C., Sitch, S., Tramontana, G., Walker, A., Weber, U., and Reichstein, M.: Scaling carbon fluxes from eddy covariance sites to globe: synthesis and evaluation of the FLUXCOM approach, Biogeosciences, 17, 1343-1365, 10.5194/bg-17-1343-2020, 2020.

Madani, N., Parazoo, N. C., Kimball, J. S., Ballantyne, A. P., Reichle, R. H., Maneta, M., Saatchi, S., Palmer, P. I., Liu, Z., and Tagesson, T.: Recent Amplified Global Gross Primary Productivity Due to Temperature Increase Is Offset by Reduced Productivity Due to Water Constraints, AGU Advances, 1, 10.1029/2020av000180, 2020.

Xiao, Z., Liang, S., and Jiang, B.: Evaluation of four long time-series global leaf area index products, Agricultural and Forest Meteorology, 246, 218-230, 10.1016/j.agrformet.2017.06.016, 2017.

Xie, X., Li, A., Jin, H., Tan, J., Wang, C., Lei, G., Zhang, Z., Bian, J., and Nan, X.: Assessment of five satellite-derived LAI datasets for GPP estimations through ecosystem models, Sci Total Environ, 690, 1120-1130, 10.1016/j.scitotenv.2019.06.516, 2019.

Yuan, W., Zheng, Y., Piao, S., Ciais, P., Lombardozzi, D., Wang, Y., Ryu, Y., Chen, G., Dong, W., Hu, Z., Jain, A. K., Jiang, C., Kato, E., Li, S., Lienert, S., Liu, S., Nabel, J. E. M. S., Qin, Z., Quine, T., Sitch, S., Smith, W. K., Wang, F., Wu, C., Xiao, Z., and Yang, S.: Increased atmospheric vapor pressure deficit reduces global vegetation growth, Science Advances, 5, 10.1126/sciadv.aax1396, 2019.